# Holding water in a sieve—stable droplets without surface tension

N. P. Longmire[1], S. L. Showalter[2] & D. T. Banuti [1,3] ✉

Our understanding of supercritical fluids has seen exciting advances over the last decades, often in direct contradiction to established textbook knowledge. Rather than being structureless, we now know that distinct supercritical liquid and gaseous states can be distinguished and that a higher order phase transition - pseudo boiling - occurs between supercritical liquid and gaseous states across the Widom line. Observed droplets and sharp interfaces at supercritical pressures are interpreted as evidence of surface tension due to phase equilibria in mixtures, given the lack of a supercritical liquid-vapor phase equilibrium in pure fluids. However, here we introduce an alternative physical mechanism that unexpectedly causes a sharpening of interfacial density gradients in absence of surface tension: thermal gradient induced interfaces (TGIIF). We show from first principles and simulations that, unlike in gases or liquids, stable droplets, bubbles, and planar interfaces can exist without surface tension. These results challenge and generalize our understanding of what droplets and phase interfaces are, and uncover yet another unexpected behavior of supercritical fluids. TGIIF provide a new physical mechanism that could be used to tailor and optimize fuel injection or heat transfer processes in high-pressure power systems.

Imagine a droplet of ink suspended in water, and rather than diffusing outwards reducing the concentration gradients, the ink drop contracts and sharpens its interface instead! This is what we found can happen in the density field under certain conditions.

In this paper, we present thermodynamic conditions under which thermal gradient-induced interfaces (TGIIF) cause the density gradient to *sharpen* instead, without any surface force acting, seemingly in contradiction to our laws of diffusion. Furthermore, this interface sharpening occurs at supercritical pressures, in contradiction to van der Waals' gradient theory[1] and classical theories of nucleation[2].

Injection at supercritical pressures is a highly relevant technology that is in widespread use in every Diesel engine[3], in every jet engine during take-off[4], and in every main-stage rocket engine[5,6]. It is essential for new emerging high-efficiency carbon-capture power cycles[7–9] and new automotive combustion cycles[10,11]. Heat transfer to supercritical

fluids is an essential technology in modern and projected power plants[12,13], including solar[14], nuclear[15], and carbon capture[14].

Over the last decade, it has become clear that a supercritical phase transition—pseudoboiling[16,17]—exists. Its main difference to subcritical boiling (or condensation) is the absence of an equilibrium coexistence of different phases; instead, the liquid–gas transition (or vice versa) occurs over a finite temperature interval[16,18,19]. Thus, despite early rejection of that idea[20], even at supercritical pressures heat transfer deterioration can be caused by a (pseudo) boiling process[21–23].

Droplet formation[24–26] at nominally supercritical pressures, on the other hand, has always been analyzed from a mixture thermodynamics standpoint. It is common for mixtures to exhibit a critical pressure that exceeds the pure fluid components' critical pressures[27]. Then, it is possible that a local mixture may find itself at a subcritical pressure, allowing phase separation, despite of the nominal supercritical pressure with respect to the pure components. This analysis has been performed

[1]Department of Mechanical Engineering, The University of New Mexico, Albuquerque, NM 87131, USA. [2]Department of Nuclear Engineering, The University of New Mexico, Albuquerque, NM 87131, USA. [3]Karlsruhe Institute of Technology (KIT), Institute for Thermal Energy Technology and Safety (ITES), Karlsruhe Institute of Technology (KIT), 76344 Eggenstein-Leopoldshafen, Germany. ✉e-mail: daniel.banuti@kit.edu

for inert[28–32] and reactive mixing[33]. On a molecular scale, supercritical phase separation in mixtures can be attributed to the interfacial thickness[34]. All existing studies focus on mixtures[28–32,35,36], where a phase equilibrium is assumed necessary for the formation of droplets.

However, the question arises as to whether the aforementioned pseudoboiling could provide a mechanism for droplets to form under true supercritical conditions, i.e., in absence of a phase equilibrium. In pseudoboiling, the liquid–gas transition occurs over a finite temperature interval across the pseudoboiling line[17], an extension of the coexistence line. It is related to the Widom line[37–40] which is defined as locus of extrema in the thermodynamic response functions and thus can have many different characteristics depending on which response function is chosen[17]. The pseudoboiling line resolves this ambiguity with a precise definition based on the curvature of the free Gibbs enthalpy[17]. It is a fluid property and approximately marks the steepest isobaric thermal density gradient $(\partial\rho/\partial T)_p$. In many injection problems, a spatial temperature gradient $(\partial T/\partial x)$ exists between the droplet and its surroundings[24–26,28–32]. Thus, the pseudoboiling line seems a likely candidate to induce a maximum in the spatial density gradient $(\partial\rho/\partial x)$, i.e., a density inflection point, in the presence of a spatial temperature gradient–much like what we see in subcritical droplets[34].

This paper introduces and analyzes an alternative mechanism that stabilizes droplets and bubbles at supercritical pressures in the absence of surface tension, based on heat transfer rather than surface forces: thermal gradient-induced interfaces (TGIIF) Specifically, we will show that, even in a pure fluid, a temperature difference between a cold/dense region and its warmer/lighter environment can be enough to cause a self-steepening and self-stabilizing density gradient that would be indistinguishable from acting surface tension in a shadowgraph.

## Results

A simplified view of a phase interface is a discontinuous transition between two fluid properties. Figure 1a illustrates this view, where at some location $\ell$ an interface marks the switch from a high density to a low density. More physically and accurately, however, is the gradual transition seen in a real vapor–liquid interface[41,42] where the position of the interface can be found in the vicinity of the density inflection point, in contrast to the density profile of a gaseous diffusion process.

**Definition 1** (Interface). Thus, here we consider an interface the location at which the density distribution exhibits a spatial inflection point.

**Definition 2** (Stable). We consider stability the tendency of some system to approach or revert to some preferred state when it is perturbed from it.

We will now demonstrate that both properties can be simultaneously present in fluids without surface tension.

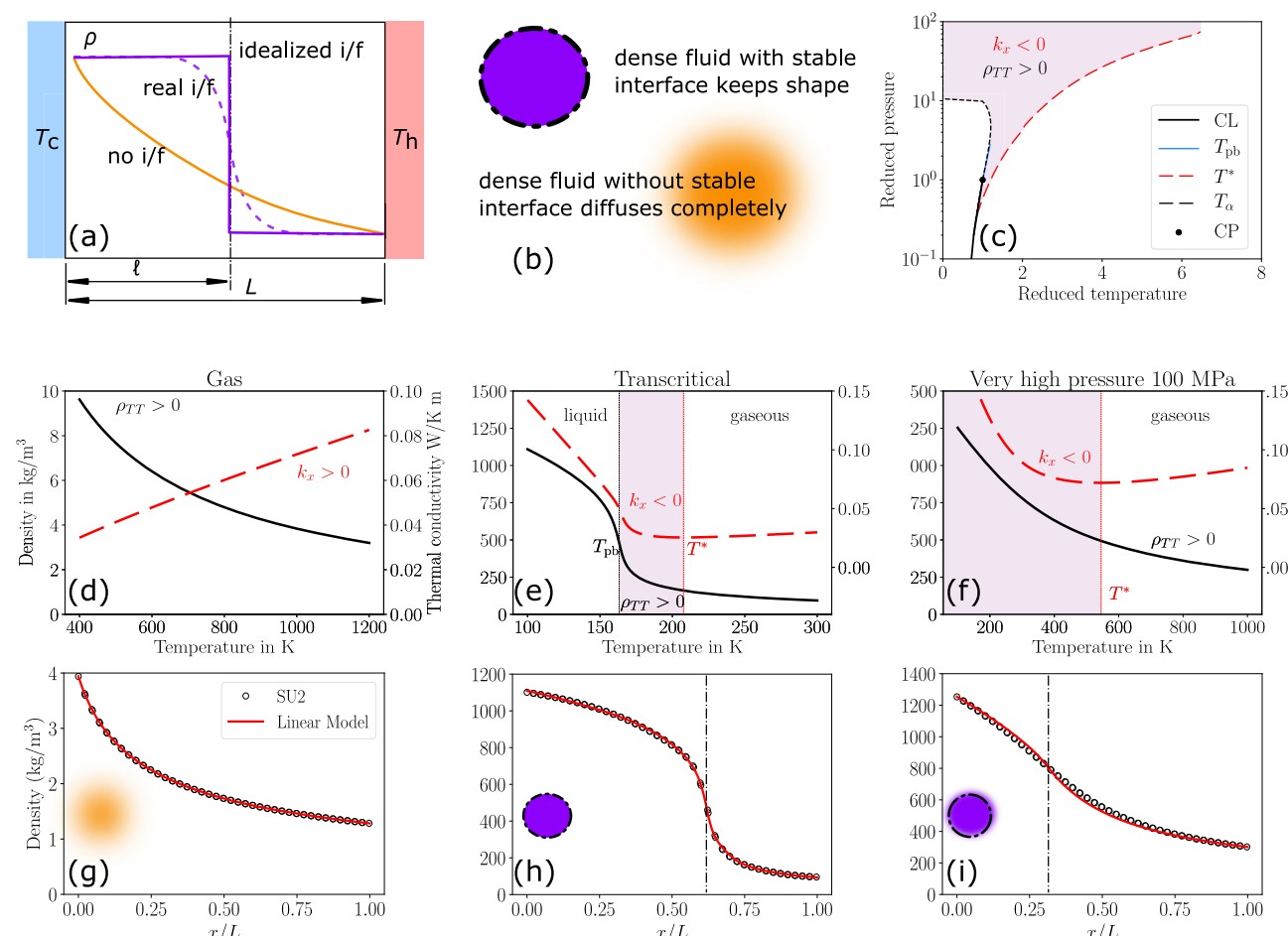

**Fig. 1 | Steady interfaces.** Top row. **a** While an idealized liquid-vapor interface is often considered discontinuous, the real interface is smooth and characterized by a density inflection point[41]. **b** With no interface, e.g., in a gas, density in the presence of a temperature gradient will drop without an inflection point. **c** Equation (9) identifies states in which a density inflection point can be caused by a temperature gradient. CL is the coexistence line, $T_{pb}$ is the pseudoboiling temperature, $T^*$ is the temperature at which the thermal conductivity reaches a minimum, $T_\alpha$ is the temperature of maximum thermal expansion. Middle row: Examples of fluid properties for **d** gas, **e** transcritical, and **f** very high-pressure fluid. The shaded area fulfils Eq. (9). Bottom: corresponding density fields at the respective conditions. Unlike the gas in (**g**), the transcritical condition (**h**) exhibits a very clear density interface. Surprisingly, even at very high pressures (**i**) where the fluid density no longer exhibits an inflection point with increasing temperature, a broad but sustained interface establishes.

## Existence of a steady interface

We start with the analysis of the steady interface at constant pressure. When the interface is characterized by an inflection point in the spatial density distribution, it is linked to the thermodynamic thermal density gradient via the spatial temperature gradient,

$$\frac{\partial \rho}{\partial x} = \frac{\partial \rho}{\partial T}\frac{\partial T}{\partial x}. \tag{1}$$

Then, using the chain rule for the 2nd derivative,

$$\frac{\partial^2 \rho}{\partial x^2} = \frac{\partial^2 \rho}{\partial T^2}\left(\frac{\partial T}{\partial x}\right)^2 + \frac{\partial \rho}{\partial T}\frac{\partial^2 T}{\partial x^2}, \tag{2}$$

setting it to zero, and using the convention of subscripts denoting differentiation, such that $\rho_T = (\partial\rho/\partial T)$ and $\rho_{TT} = (\partial^2\rho/\partial T^2)$, yields

$$-\frac{\rho_{TT}}{\rho_T} = \frac{1}{\left(T_x\right)^2}T_{xx}. \tag{3}$$

Using Fourier's law of heat conduction in 1D

$$q'' = k\frac{\partial T}{\partial x} = kT_x \tag{4}$$

to eliminate $T_x$ results in the steady-state relation

$$-\frac{\rho_{TT}}{\rho_T} = \left(\frac{k}{q''}\right)^2 T_{xx}. \tag{5}$$

Equation (5) needs to be fulfilled for a density inflection point, and thus interface, to exist. The RHS describes heat transfer characteristics, whereas the LHS represents fluid thermodynamic properties.

The simplest way to analyze Eq. (5) is to check for sign consistency. The squared term on the RHS is always positive. This means that an inflection point for fluids that expand when heated, i.e., $\rho_T < 0$, can exist if and only if

$$\mathrm{sgn}(\rho_{TT}) = \mathrm{sgn}\left(T_{xx}\right). \tag{6}$$

Further, through differentiation of Eq. (4) for steady state with constant $q''$ throughout the field,

$$q''_x = kT_{xx} + k_x T_x = 0, \tag{7}$$

where $k > 0$. Without loss of generality, we regard the case of Fig. 1a with a positive temperature gradient $T_x > 0$ and find

$$\mathrm{sgn}(\rho_{TT}) = -\mathrm{sgn}(k_x). \tag{8}$$

Finally, with $k_x = (\partial k/\partial T)(\partial T/\partial x)$, we obtain

$$\mathrm{sgn}(\rho_{TT}) = -\mathrm{sgn}(k_T) \tag{9}$$

as a necessary condition for the existence of an interface. Note that Eq. (9) is a criterion based entirely on fluid properties, and does not depend on experimental boundary conditions. Figure 1c shows the region in which Eq. (9) is fulfilled in a pressure-temperature diagram.

## Fluid state case evaluations

Using Eq. (9), we can now evaluate different fluid states for their potential to develop interfaces, results are shown in Fig. 1. For simplicity, we will restrict ourselves to isobaric cases, it is instructive to view the fluid property diagrams in "Methods". Here, $T_{pb}$ is the pseudoboiling temperature[16] which characterizes the supercritical liquid–gas transition and is defined as the locus of the isobaric heat capacity peaks. $k_{min}$ is the minimum value of the thermal conductivity evaluated at constant pressure and $T^*$ is the temperature at which $k_{min}$ is reached.

In an ideal gas with $\rho = p/RT$, $\mathrm{sgn}(\rho_{TT}) > 0$, kinetic theory suggests a dependence $k \propto \sqrt{T}$, thus $\mathrm{sgn}(k_T) > 0$. As this contradicts Eq. (9), an ideal gas cannot exhibit an interface.

The same analysis can be used to show that interfaces cannot occur in liquids with typically $\mathrm{sgn}(\rho_{TT}) < 0$ and $\mathrm{sgn}(k_T) < 0$.

In a transcritical fluid, for $T < T_{pb}$, $\mathrm{sgn}(\rho_{TT}) < 0$ and $\mathrm{sgn}(k_T) < 0$ like in a liquid, preventing an interface. On the other hand, for $T > T^*$, with $T^* := T(k_{min}) = T^*$, $\mathrm{sgn}(\rho_{TT}) < 0$ and $\mathrm{sgn}(k_T) > 0$, like in a gas, likewise preventing an interface. This leaves $T_{pb} < T < T^*$ as a region of unique fluid properties with $\mathrm{sgn}(\rho_{TT}) > 0$ and $\mathrm{sgn}(k_T) < 0$, allowing for an interface to form.

Figure 1f shows in a $p$-$T$ diagram that an interface can exist at high subcritical and even very high supercritical pressures. In fact, we did not find an upper pressure limit to the existence of a density inflection point.

## Analytical analysis and steady 1D simulation

A supercritical steady interface can be analyzed analytically. Consider two parallel walls at different temperatures, a cold $T_c$ and a hot $T_h$. The space between them is filled with a pure fluid and the pressure is kept constant. The fluid does not flow, so heat is only transferred by conduction. This will allow our heat transfer to be analyzed by Fourier's law, Eq. (4). In fact, the temperature distribution is solely governed by the boundary temperatures and the fluid thermal conductivity.

An accurate modeling of transcritical fluid properties is an ongoing problem[43,44]. Here, we approximate the complex thermal conductivity distribution shown in Fig. 1 as piece-wise linear, with a liquid branch and a gaseous branch, a detailed derivation can be found in the methodology. Then, exact analytical solutions to the temperature distribution can be determined from Fourier's law,

$$T(x) = \frac{-b_c \pm \sqrt{b_c^2 - 2m_c\left[q''(x-x_0) - \frac{m_c}{2}T_c^2 - b_c T_c\right]}}{m_c} \quad \text{when } x \le \hat{x} \tag{10}$$

$$T(x) = \frac{b_h \pm \sqrt{b_h^2 + 2m_h\left[-q''(x-x_h) + \frac{m_h}{2}T_h^2 + b_h T_h\right]}}{-m_h} \quad \text{when } x > \hat{x} \tag{11}$$

Figure 1 compares the analytical solution to computational fluid dynamics simulations using neural-network fit equations for fluid properties[4,22,45] which we implemented in the open-source solver SU2.

Somewhat unexpectedly, agreement between the analytical and the numerical solutions is excellent for a wide range of conditions. Beginning with the ideal gas, the linear model closely matches the density distribution of the simulation, which exhibits no inflection point and thus no interface. At transcritical conditions (7 MPa, $p/p_{cr} \approx 1.2$), a very distinct inflection point develops, separating the domain into a liquid-filled and a gas-filled side. Even at extremely high pressures of 100 MPa ($p/p_{cr} \approx 20$) does the inflection point not vanish; however, the density gradient has become very smooth and a clear separation into liquid and gaseous has all but vanished.

## Transient gradient sharpening

To study the transient interface, we perform a 2D parametric study looking at droplets of n-heptane at several different pressures and environmental temperatures.

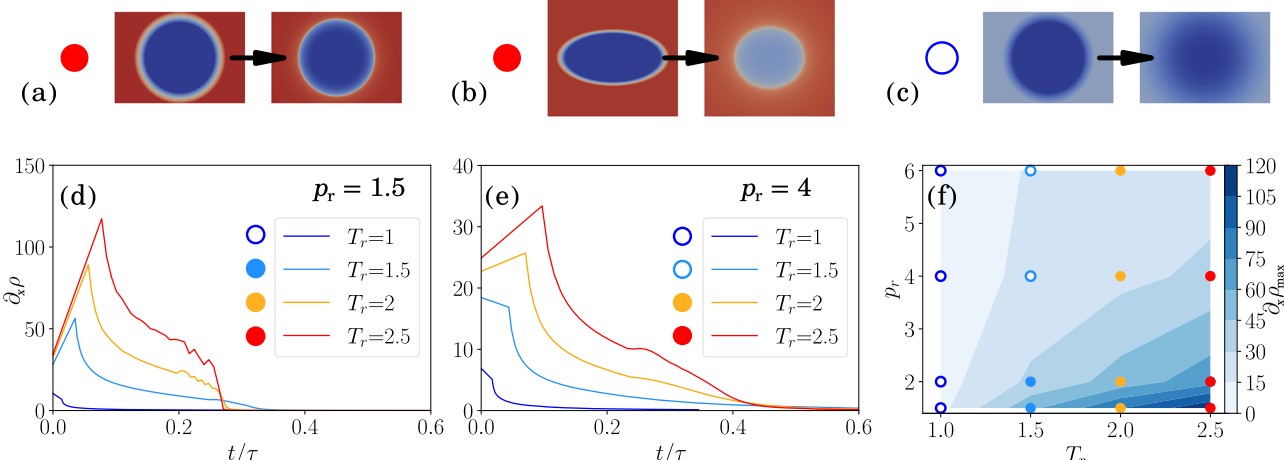

**Fig. 2 | Transient interfaces.** Filled circles mark TGIIF, empty circles mark diffuse interfaces. Examples: **a** An initially broad interface sharpens over time; **b** an elliptical cross-section regresses towards a circular shape; **c** when the temperature gradient is too low, the droplet diffuses. **d**, **e** The maximum spatial density gradient $\partial_x\rho$ increases linearly if the surroundings are hot enough, leading to TGIIF. At some point, the density gradient reduces and droplets vaporize. For higher pressures, a higher $\Delta T$ is required to cause TGIIF and the magnitude of the density gradients reduce. **f** Occurrence of TGIIF and diffuse interfaces in a pressure-temperature diagram. The background contour plot shows the maximum spatial density gradient, the solid dots show the formation of stable TGIIF, and the non-filled points represent diffusive interfaces. The colors of the points correspond to the farfield reduced temperatures, consistent with (**d**) and (**e**).

The parametric study mirrors experimental studies of Diesel injection[25,26], in which hydrocarbon droplets at $T_d = 363$ K are observed in hot high-pressure environments. Here, we embed the supercritical liquid $n$-heptane droplet in warm supercritical gaseous environments at different temperatures $T_r = [1, 1.5, 2, 2.5]$, pressures $p_r = [1.5, 2, 4, 6]$ and diameters $D = [0.001, 0.01, 0.1]$ m. The initial interface between liquid and gas follows a linear density profile.

Figure 2 demonstrates that TGIIF are indeed acting to create stable interfaces: Fig. 2a illustrates how an initially fuzzy droplet develops a sharper interface over time; Fig. 2b demonstrates the regression of an elliptical cross-section toward a circular cross-section; finally, Fig. 2c shows a supercritical droplet without TGIIF, diffusing, as one would expect for all of them.

In order to analyze the vaporization transient, we determine a representative timescale from dimensional analysis of transient heat conduction, c.f. "Time scale analysis",

$$\frac{\partial T}{\partial t} = \alpha \frac{\partial^2 T}{\partial x^2}. \tag{12}$$

The associated timescale is then

$$\tau \propto \frac{\theta}{\Delta T}\frac{D^2}{\alpha}. \tag{13}$$

We define the nondimensional time $\tau$ where $\alpha$ is the minimum thermal diffusivity for a fluid at a given pressure, $\theta$ is the temperature at which the minimum thermal diffusivity occurs, $\Delta T$ is the temperature between the warm gas and the cold liquid, and $D$ is the initial diameter of the droplet.

The transient development of the maximum density gradient is quantified in Fig. 2a and b for two different pressures. We can clearly see how the density gradient grows linearly until a maximum is reached, if the temperature difference $\Delta T$ between droplet and surroundings is large enough. Figure 2f maps the positions at which TGIIF and diffuse interfaces are found.

**Transient vaporization**

The dimensional analysis in Eq. (13) suggests that supercritical vaporization will be linear in $D^2$, akin to Spalding's $D^2$ law in subcritical droplet evaporation[46]. Following Fig. 2, we consider the interface of the droplet the position of the maximum density gradient, then $D$ is the diameter based on that location. Figure 3a−c shows how the square of the diameter $D$, nondimensionalized with the initial diameter $D_0$, indeed shrinks linearly in time for TGIIF conditions−much like a subcritical droplet! For sufficiently high $\Delta T$, the curves collapse; for $T_r < 2$ and $p_r < 4$, the droplet grows initially before following the $D^2$ relation.

Figure 3d, e reveals another unexpected similarity of TGIIF to subcritical droplet evaporation: Regardless of the environmental temperature, the temperature at the interface matches a single temperature−the pseudoboiling temperature $T_{pb}$, which can be considered the supercritical complement to subcritical saturation temperature, and is found on the pseudoboiling line as an extension to the coexistence line[16,17]. Pseudoboiling seizes to be a relevant phenomenon for $p_r > 3$, and Fig. 3f shows a larger deviation from $T_{pb}$, although it still appears to act as a (weak) attractor.

We now turn to the droplet lifetimes as our final analysis. Figure. 3g, h illustrates the droplet evolution using snapshots at different stages of vaporization, for a stable TGIIF droplet and a diffuse droplet, respectively. We consider the droplet lifetime the time until the maximum density gradient has reached its minimum value or a zero radius. We can then compile and compare droplet lifetimes over the whole parameter range, i.e., $T_r = [1, 1.5, 2, 2.5]$, $p_r = [1.5, 2, 4, 6]$, $D = [0.001, 0.01, 0.1]$ for all cases that exhibited TGIIF.

A dimensional analysis, such as the one performed to obtain Eq. (13) and as presented in more detail in the methods part, can reveal relations only up to a proportionality constant. We thus introduce a vaporization number Va as the nondimensional proportionality constant to Eq. (13),

$$Va = t\frac{\Delta T}{\theta}\frac{\alpha}{D^2}. \tag{14}$$

We find that for Va = 0.31 the predicted droplet lifetime is accurate to within 50% over five decades of lifetimes obtained from simulations. Figure 3i illustrates this. Thus, Va = 0.31 seems to be the characteristic vaporization number for the problem at hand.

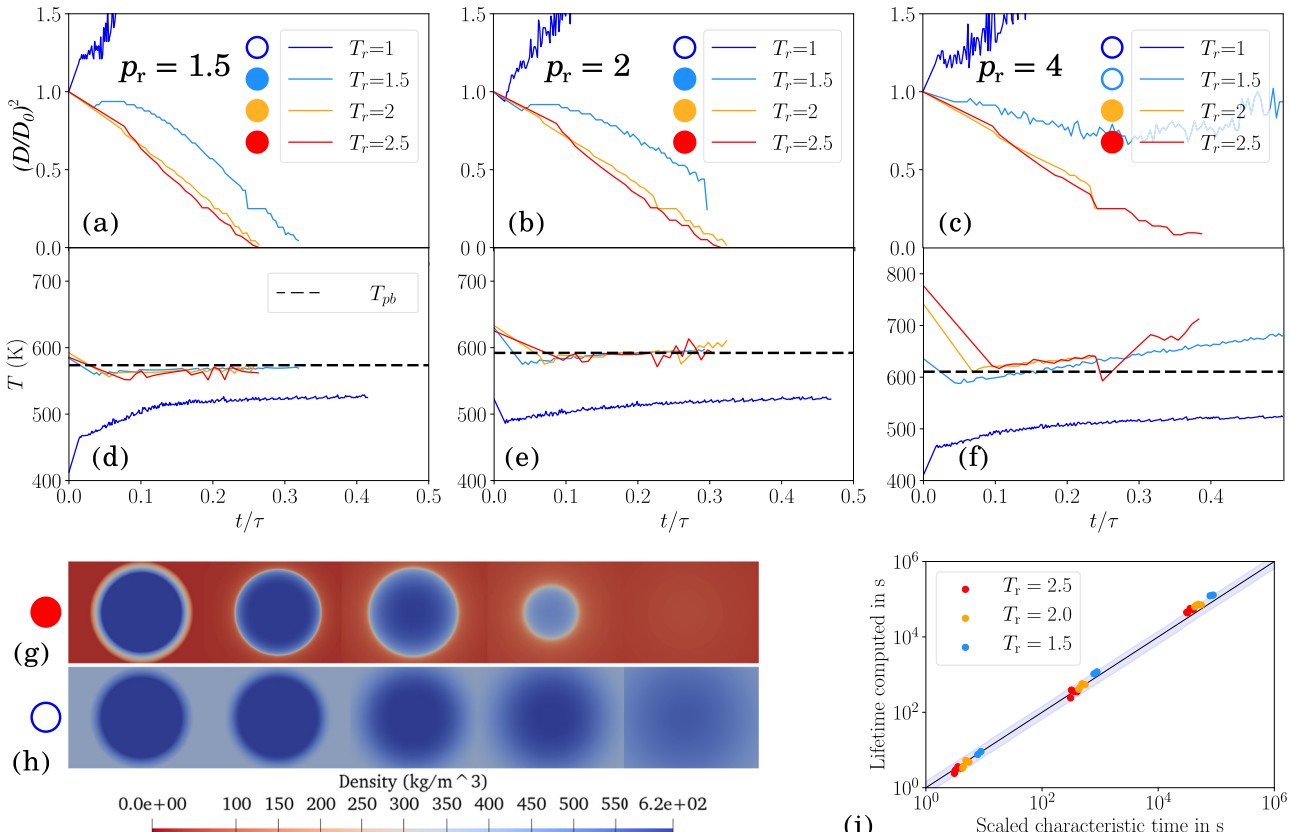

**Fig. 3 | Supercritical droplet vaporization. a–c** Droplet size decreases linearly in $D^2$ for TGIIF, like during subcritical evaporation. **d–f** The interfacial temperature matches the pseudoboiling temperature $T_{pb}$ with high accuracy. **g, h** Snapshots of the evolution of density contour for TGIIF and diffuse droplet. TGIIF droplet snapshots timestamps from left to right $t/\tau = [0, 0.049, 0.136, 0.262, 0.298]$ and diffusive droplet snapshots timestamps from left to right $t/\tau = [0, 0.087, 0.192,$ 0.309, 0.334] **(i)** Equation (14) matches simulated droplet lifetimes over five decades over the whole parameter range, i.e., $T_r = [1, 1.5, 2, 2.5]$, $p_r = [1.5, 2, 4, 6]$, $D = [0.001, 0.01, 0.1]$ for all cases that exhibited TGIIF. Shaded region marks 50% error. 'Lifetime computed' using CFD, c.f. **a–c**; "scaled characteristic time" calculated using Eq. (14).

## Discussion

We introduce a new physical mechanism that acts to stabilize liquid–gas interfaces, such as droplets or bubbles, in the absence of phase equilibrium and surface tension: thermal gradient-induced interfaces (TGIIF).

TGIIF are surprisingly similar to classical droplet evaporation: TGIIF droplets exhibit sharpening density gradients and cause elliptical cross-sections to regress towards circular cross-sections; they vaporize following a $D^2$ law, while their interfaces hold the supercritical pseudoboiling temperature, akin to the subcritical saturation temperature.

Our first principle analysis of criteria for a stable density inflection point yields as a necessary condition that the concavity of the thermal density gradient and the slope of the thermal conductivity have opposite signs, $\mathrm{sgn}(\rho_{TT}) = -\mathrm{sgn}(k_T)$. This condition holds under transcritical conditions; it does not hold in liquids or gases.

The nondimensional vaporization number Va relates droplet size, temperature gradients, and thermal diffusivity to droplet lifetime. We find that Va = 0.31 matches simulated droplet lifetimes over five decades, within 50% error.

Ultimately, this work questions and extends the definition of what constitutes a droplet or a bubble. Photographic evidence of stable spherical structures in high-pressure injection does not necessarily imply the existence of a phase equilibrium or surface tension. Our study answers the long standing question as to whether transcritical interfaces can be stable: they can, even in absence of surface tension. It provides justification for two-layer models for supercritical heat transfer analysis.

## Methods
### Solver
The CFD solver used was open-source SU2[47,48], more specifically the low-mach approximation solver was used. The convection scheme used is the flux difference splitting scheme (FDS). The FDS scheme is an upwind scheme and typically has first-order accuracy, but second-order accuracy is achieved via Monotonic Upwind Scheme for Conservation Laws (MUSCL)[47]. The solver was then extended with tiny neural networks (TNN) in order to model efficiently and accurately capture the complex supercritical fluid properties. An in-depth explanation of the TNN and its implementation into SU2 are found in refs. 22,45, representative fluid property distributions are shown for n-heptane and oxygen in Fig. 4.

The case was set up so the left half of the domain was the cold liquid side with cold wall of the same temperature on the edge, and on the right side was the warm vapor with a hot wall of the same temperature on the edge. The heat would then diffuse from the warm side to the cold side until a constant heat flux was reached across the domain and that was the steady state.

### Analytical model methodology
A supercritical droplet or bubble structure should be described analytically. To portray a supercritical droplet, Fourier's Law is used across a simple one-dimensional steady-state case with a linear approximation of the fluid thermal conductivity. The linear model allows for hand derivations of an equation to describe

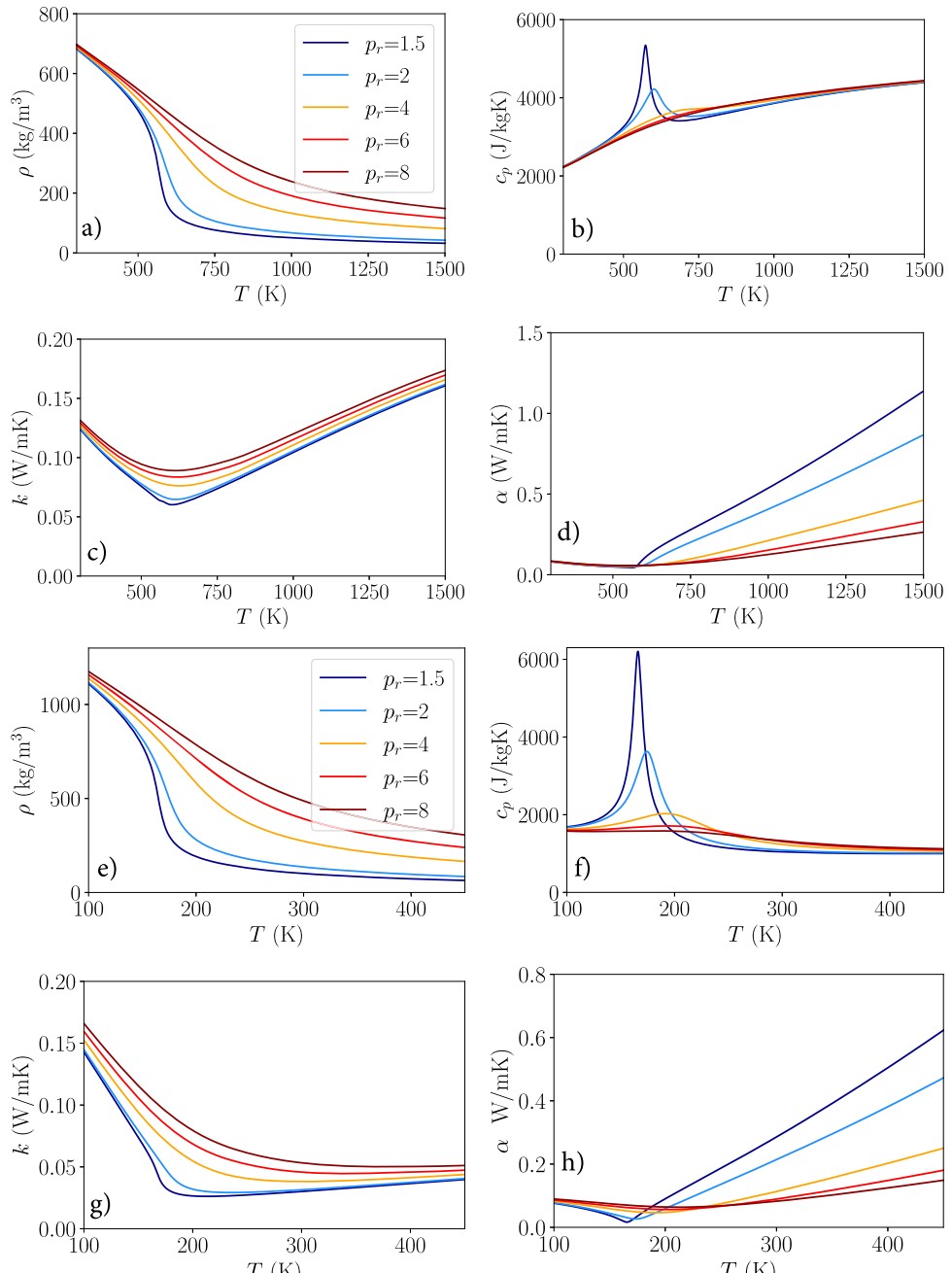

**Fig. 4 | Isobaric fluid properties at different reduced pressures ($p_r$) for $n$-heptane and oxygen. a–d** Fluid properties for $n$-heptane. **e–h** Fluid properties for oxygen. **a**, **e** Density ($\rho$). **b**, **f** Isobaric heat capacity ($c_p$). **c**, **g** Thermal conductivity ($k$). **d**, **h** Thermal diffusivity ($\alpha$).

the temperature profile which is then coded in Python to iterate across a spatial domain and plot the steady-state temperature profile.

**Deriving a pseudoboiling distance.** The analytical case is defined as a cold wall of temperature $T_c$ and a hot wall of temperature $T_h$ separated by a distance L. The steady-state heat flux is governed by Fourier's Law.

$$q'' = -k\nabla T \qquad (15)$$

For a 1D model, the gradient is equal to d$T$/d$x$. Thermal conductivity is not constant and can be expressed as a function of temperature.

$$-q'' = k(T)\frac{dT}{dx} \qquad (16)$$

Next, a fluid will be chosen at a pressure which pseudoboiling occurs. Temperatures $T_c$ and $T_h$ are then defined as below and above the pseudoboiling temperature $T_{pb}$, respectively. This is intended to capture the pseudoboiling transition. A linear function is then fit to the thermal conductivity such that $k_c(T)$ spans from $T_c$ to some arbitrary interface temperature, $\hat{T}$. This region will be referred to as the cold side. Also, a linear function $k_h(T)$ approximates the region from $\hat{T}$ to $T_h$, the hot side.

$$k_c(T) = m_c(T) + b_c \qquad (17)$$

To create a linear equation for each region, a temperature between the wall and interface temperatures is chosen $\bar{T}_{\text{region}}$ for both the hot and cold sides. These line-fitting temperatures have a corresponding thermal conductivity, $\bar{k}_{\text{region}}$.

$$m_{\text{region}} = \frac{\bar{k}_{\text{region}} - k_{\text{wall}}}{\bar{T}_{\text{region}} - T_{\text{wall}}} \tag{18}$$

and

$$b_{\text{region}} = k_{\text{wall}} - m_{\text{region}} T_{\text{wall}} \tag{19}$$

In this model, there exists an $\hat{x}$ where $\hat{T}$ occurs. Because the model is in a steady state, the heat flux is constant across any region $dx$. So, the heat flux across the region below the transition temperature shall equal the heat flux in the region above the transition temperature and $k(T)$ can be substituted in

$$\frac{(m_c(T) + b_c)dT}{dx_c} = \frac{(m_h(T) + b_h)dT}{dx_h}. \tag{20}$$

Next, both sides are integrated over respective temperatures and distances,

$$\frac{\int_{T_c}^{\hat{T}} (m_c(T) + b_c)dT}{\int_{x_c}^{\hat{x}} dx} = \frac{\int_{\hat{x}}^{x_h} (m_h(T) + b_h)dT}{\int_{\hat{x}}^{x_h} dx_h}, \tag{21}$$

and the integral is solved

$$\frac{(m_c/2)\left(\hat{T}^2 - T_c^2\right) + b_c(\hat{T} - T_c)}{\hat{x} - x_c} = \frac{(m_h/2)\left(T_h^2 - \hat{T}^2\right) + b_h(T_h - \hat{T})}{x_h - \hat{x}}. \tag{22}$$

The equations can be rearranged and the large ratio of integrated $k(T)$ $dT$ represented as $\xi$.

$$\frac{x_h - \hat{x}}{\hat{x} - x_c} = \frac{(m_h/2)\left(T_h^2 - \hat{T}^2\right) + b_h\left(T_h - \hat{T}\right)}{(m_c/2)\left(\hat{T}^2 - T_c^2\right) + b_c\left(\hat{T} - T_c\right)} = \xi \tag{23}$$

Rearranged with $\xi$

$$x_h - \hat{x} = \xi\hat{x} - \xi x_c \tag{24}$$

And rearranged again to isolate $\hat{x}$

$$\hat{x} = \frac{x_h - \xi x_c}{\xi + 1} \tag{25}$$

With (25), $\hat{x}$ is known for a fully defined case with: constant heat flux; a particular isobaric fluid; hot, cold, and pseudoboiling temperatures; and a discrete region $x_c$ to $x_h$.

**Plotting a temperature profile.** To create a temperature profile, Fourier's Law (16) can be integrated across the domain, 0 to $x$, and from $T_c$ to $T(x)$ (26). Because $k(T)$ is different across the two regions, the results will differ depending on what value of $x$ is chosen.

$$\int_{x_0}^{x} q'' dx = \int_{T_c}^{T(x)} -k(T)dT \tag{26}$$

For $x <= \hat{x}$ the integration results in (27) where $T(x)$ can be solved for with the quadratic formula, and positive results chosen in (28).

$$-q''(x - x_0) = \frac{m_c}{2}\left[T(x)^2 - T_c^2\right] + b_c(T(x) - T_c) \tag{27}$$

$$T(x) = \frac{-b_c \pm \sqrt{b_c^2 - 2m_c\left[q''(x - x_0) - \frac{m_c}{2}T_c^2 - b_c T_c\right]}}{m_c} \quad \text{when} \ x \leq \hat{x} \tag{28}$$

For $x > \hat{x}$, we can integrate over the hot region, which yields in a similar formula

$$T(x) = \frac{b_h \pm \sqrt{b_h^2 + 2m_h\left[-q''(x - x_h) + \frac{m_h}{2}T_h^2 + b_h T_h\right]}}{-m_h} \quad \text{when} \ x > \hat{x} \tag{29}$$

With an expression for $T(x)$, the spatial temperature gradient can be found as well by finding the derivative of $T(x)$ in respect to x. This yields the following two equations.

$$\frac{dT(x)}{dx} = \frac{-q''}{\sqrt{b_c^2 - 2m_c\left[q''(x - x_0) - (m_c/2)T_c^2 - b_c T_c\right]}} \quad \text{if} \ x \leq \hat{x} \tag{30}$$

$$\frac{dT(x)}{dx} = \frac{-q''}{\sqrt{b_h^2 - m_h\left[m_h T_h^2 + 2T_h b_h + 2q''(x - x_h)\right]}} \quad \text{if} \ x > \hat{x} \tag{31}$$

## 1D steady state

The analytical model outlined in "Analytical model methodology" can be applied to oxygen at 7 MPa between 100 K and 300 K walls, across a unit domain of 1 m to yield fractional results. For comparison to a computational solver, SU2, the relative domain results are applied to a 1-mm distance between hot and cold walls.

To best represent the thermal conductivity of 7-MPa oxygen, the equations for the thermal conductivity model (k-model) were chosen to reflect the liquid region between 100 K and 158 K for the cold side, and between 250 K and 300 K to reflect the gaseous region. The resulting k-model is shown below in Fig. 5 and a sample of cases with parameters of interest can be found in Table 1.

## 2D droplets and bubbles

**Mesh.** In Figure 6 are pictures of the mesh used for the 2D droplet simulations. The left picture shows the whole domain and the right picture zooms in on the center of the mesh to show the refined part of the mesh where the interface occurs.

## Time-scale analysis

We identify a representative timescale of the vaporization process by performing dimensional analysis. 1D heat conduction through a constant property medium is governed by

$$\frac{\partial T}{\partial t} = \alpha \frac{\partial^2 T}{\partial x^2}. \tag{32}$$

We define reference values and corresponding nondimensional variables such that $\bar{x} = x/D$ and $\bar{t} = t/\tau$. For the temperature term on the lhs we introduce a reference temperature $\bar{T} = T/\theta$, on the rhs we choose a reference temperature difference instead, $\bar{T} = T/\Delta T$. Then,

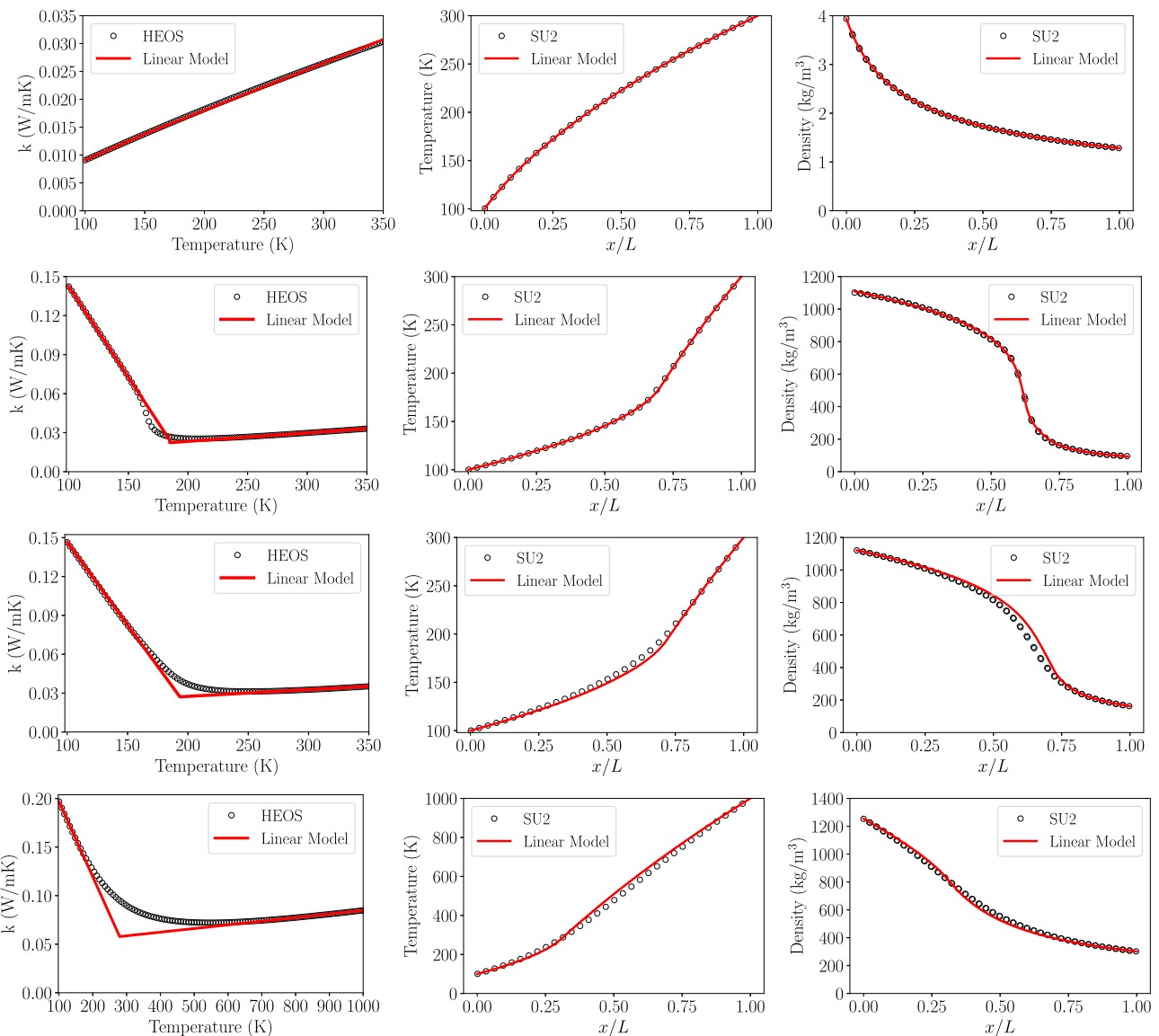

**Fig. 5 | A set of linear model results compared with SU2 simulations for thermal conductivity, temperature profile, and density profiles.** The cases of rows are from top to bottom: oxygen at 0.1 MPa (ideal gas), 7 MPa, 12 MPa, and 100 MPa.

Eq. (32) can be rewritten

$$\frac{\partial \bar{T}}{\partial \bar{t}} = \frac{\theta}{\Delta T} \frac{\tau \alpha}{D^2} \frac{\partial^2 \bar{T}}{\partial \bar{x}^2}, \tag{33}$$

and we identify a nondimensional group which we will dub the vaporization number

$$\text{Va} = \frac{\theta}{\Delta T} \frac{\tau \alpha}{D^2}. \tag{34}$$

**Table 1 | Linear model parameters for oxygen**

| Pressure (MPa) | $m_c$ (W/mK/K) | $b_c$ (W/mK) | $m_h$ (W/mK/K) | $b_h$ (W/mK) | $T_{tr}$ (K) |
|---|---|---|---|---|---|
| 0.1 | −9.391E-5 | 3.06E-4 | 8.470E-5 | 1.077E-3 | 150.00 |
| 7 | −1.394E-3 | 0.282 | 6.513E-5 | 0.01030 | 184.70 |
| 12 | −1.279E-3 | 0.274 | 5.166E-5 | 0.01715 | 193.21 |
| 100 | −7.751E-4 | 0.274 | 3.746E-5 | 0.04740 | 279.50 |

The associated timescale is then

$$\tau = \text{Va} \frac{\theta}{\Delta T} \frac{D^2}{\alpha}. \tag{35}$$

$\tau$ is used in our analysis to nondimensionalize the time. $\alpha$ is the minimum thermal diffusivity for a fluid at a given pressure, $\theta$ is the temperature at which the minimum thermal diffusivity occurs, $\Delta T$ is the temperature between the warm gas and the cold liquid, and $D$ is the initial diameter of the droplet.

## Data availability
The fluid property data used for this study was through Coolprop[49] and NIST[50], and the data is available through their websites. The data can be requested from the authors.

## Code availability
The tiny neural networks used for fluid property modeling is the only custom code that was used and it is available through the GitLab website https://gitlab.com/tfxlab/thermoml.git. The solver used for

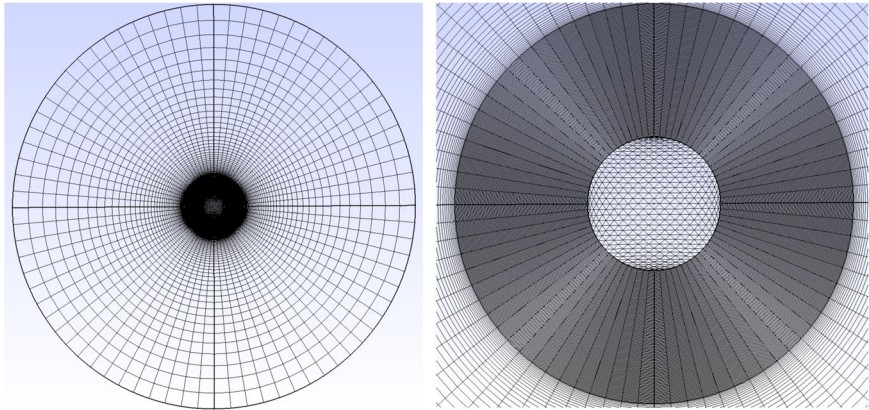

**Fig. 6 | Pictures of mesh used for droplet simulations.** Left is an image of the whole domain, and the right is an image zoomed into the center where the interface is located.

the study, open-source SU2, is available on their GitHub website https://github.com/su2code/SU2.

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

## Acknowledgements

We gratefully acknowledge the support of AFRL/RBK through DEPSCoR FA9550-22-1-0306, which funded this research. D.T.B.

## Author contributions

Conceptualization: D.T.B. and N.P.L.; methodology: D.T.B.; software: N.P.L. and D.T.B.; validation: N.P.L.; resources: D.T.B.; data curation: N.P.L. and S.L.W.; writing—original draft preparation: N.P.L., D.T.B., and S.L.W.; writing—review and editing: D.T.B.; visualization: N.P.L. and S.L.W; supervision: D.T.B.; project administration: D.T.B.; funding acquisition: D.T.B.

## Competing interests

The authors declare no competing interests.
