## [Peer Review File · Nature Communications]

REVIEWER COMMENTS

Reviewer #1 (Remarks to the Author):

The authors present in this article a new mechanism occurring in a supercritical fluid medium, for particular temperatures and pressures, thus highlighting the appearance of a pseudo interface under the action of a strong temperature gradient. This mechanism is made possible by the particular properties of the supercritical fluid from liquid to gas, especially for conditions close to the "Widom line". If the concept of pseudo boiling is not new in such media, the authors demonstrate in a simple and smart way that a pseudo interface can exist, without surface force, but under the action of a strong temperature gradient. A first result concerns the establishment of a relation imposing the presence of an inflection point in the spatial distribution of the density (definition of the location of the interface) involving the second derivatives of the density and the temperature. This relation allows to determine the zone in the pressure-temperature diagram for which this type of stable interface appears. Subsequently, a convincing analysis, both analytical and numerical, is proposed to support the concept of stable interface without surface tension. The proposal of a characteristic time of the phenomenon and the definition of a pseudo evaporation temperature are quite relevant. The authors also show that the temperature of the pseudo interface tends to a constant value in the same way as the saturation temperature in subcritical conditions. This reinforces the concept of pseudo boiling line.

I really appreciated this work and I think that this contribution is quite original. It could be improved by taking into account the following remarks

I equation 7, clarify what is kT and give more explicit explanation to understand the link between eq 6 and eq 7 ? maybe eq 14 can be added in the main text.

can you add a scheme of the numerical configuration

can you precise more the fact that an interface can exist at high pressures ? What do you want to say "there is no upper limit " 194 p5 ?

what is the order of magnitude of the vaporization transient timescale ?

in figure 3g and h, can you precise the time for each image ?

I think the figure 2f has to be more described for an ease of understanding.

The relation 12 and figure 3i should be more detailed, why $V_a = 0.31$?

Reviewer #2 (Remarks to the Author):

The work is devoted to the analysis of supercritical fluids in the presence of temperature gradients inside the system. Both analytical results and results on computer simulation of real fluids were obtained. It is concluded that in the presence of temperature gradients, an analog of surface tension will arise in the system. The equilibrium distribution of the order parameter (eg density deviation from the mean) is given by the temperature distribution, hence its gradient.

Hence, surface tension automatically arises. The results themselves are trivial and do not correspond to the level of the journal. Moreover, the situation with temperature gradients is, by definition, nonequilibrium. No estimates of the characteristic resorption times for temperature and density gradients are given. If we perform a detailed analysis of the kinetics of processes and characteristic times for real fluids, then the article could be published in a technical journal.

Reviewer #3 (Remarks to the Author):

This paper highlights a very interesting research with is intriguing with unusual conclusions. The conclusion of the authors is explicit and illustrate my purpose. They say: "Ultimately, this work questions and extends the definition of what constitutes a droplet or a bubble". Yes, this is a good question.

The main idea is to consider some cases with a temperature gradient across the Widom line (between supercritical liquid and gaseous states). They call it "thermal gradient induced interfaces (TGIIIF)". The interface is defined when the density has an inflection point. The mathematical and thermodynamical concepts are classical (Fourier law, simple algebra) even if some parts are derived without some clear explanation. The paper is of good quality even if one the authors is a specialist of this topic as he has been publishing on Widom line for quite a few years. The idea is very interesting even if it is unexpected. I think the paper can be published in "Nature&Communications" after having answered to my following questions/remarks:

1) Is it appropriate to call it "droplet"? We are in fact in liquid and gas like regions but without any surface tension. I am not sure if it is appropriate to call it "droplet".

2) The authors state the supercritical vaporization is analogous to subcritical droplet evaporation and in the conclusion they are talking about multiphase interfaces which troubles me a little bit. A question related to these facts: what would be the result of a 2D or 3D modelling? I guess the curvature might play a non-negligible role.

3) I guess the time effect is important in figure 2 as clearly shown by the authors. The sharp interface scenario does not stay forever. Would one consider that this resembles to a real interface with surface tension? I do not think so.

4) Eqs. (1-7) are clear but the explanation which follows on page 5 is not easy to follow as some parameters are not clearly defined for the transcritical case. What does mean " ρb "? How to define k_{min} ? What is the value of T^* ? This part needs to be more developed.

5) There is a mistake on page 4 and 5 (Eq. (14) should be Eq. (4)).

Response to Reviewers

We would like to thank the reviewers for the time and the valuable comments. We are excited to read that R1 “really appreciated this work” and finds “that this contribution is quite original”, and that R2 finds “the idea is very interesting” and “unexpected”. Without doubt, addressing the comments has helped make the paper much clearer. Thank you!

Reviewer 1:

The authors present in this article a new mechanism occurring in a supercritical fluid medium, for particular temperatures and pressures, thus highlighting the appearance of a pseudo interface under the action of a strong temperature gradient. This mechanism is made possible by the particular properties of the supercritical fluid from liquid to gas, especially for conditions close to the "Widom line". If the concept of pseudo boiling is not new in such media, the authors demonstrate in a simple and smart way that a pseudo interface can exist, without surface force, but under the action of a strong temperature gradient. A first result concerns the establishment of a relation imposing the presence of an inflection point in the spatial distribution of the density (definition of the location of the interface) involving the second derivatives of the density and the temperature. This relation allows to determine the zone in the pressure-temperature diagram for which this type of stable interface appears. Subsequently, a convincing analysis, both analytical and numerical, is proposed to support the concept of stable interface without surface tension. The proposal of a characteristic time of the phenomenon and the definition of a pseudo evaporation temperature are quite relevant. The authors also show that the temperature of the pseudo interface tends to a constant value in the same way as the saturation temperature in subcritical conditions. This reinforces the concept of pseudo boiling line.

I really appreciated this work and I think that this contribution is quite original. It could be improved by taking into account the following remarks:

A: We would like to thank the reviewer for the comments and the assessment that this work is quite original and that it introduces a simple and smart way that a pseudo interface can exist without surface force. We are positive we have addressed each remark in the revised manuscript.

R1.1: In equation 7, clarify what is kT and give more explicit explanation to understand the link between eq 6 and eq 7 ? maybe eq 14 can be added in the main text.

A1.1: Thank you for pointing this out. We extended this discussion and hope it is clearer now.

R1.2: Can you add a scheme of the numerical configuration

A1.2: Thank you for the suggestion. The convection scheme used is the flux difference splitting scheme (FDS). The FDS scheme is an upwind scheme and typically has first-order accuracy, but second-order accuracy is achieved via Monotonic Upwind Scheme for Conservation Laws (MUSCL). The methods section in the manuscript has been updated to include this information.

R1.3: Can you precise more the fact that an interface can exist at high pressures ? What do you want to say “there is no upper limit “ I 94 p5 ?

A1.3: Thank you for the comment. The sentence in the manuscript has been corrected to refer to the correct figure and has been rewritten to clarify the statement. “ Figure~\ref{fig:steady-fig}f shows in a p-T diagram that an interface can exist at high subcritical and even very high supercritical pressures. In fact, we did not find an upper pressure limit to the existence of a density inflection point.” We were somewhat surprised by this finding as we expected an upper pressure limit comparable to pseudo boiling, i.e. for $pr > 3$.

R1.4: what is the order of magnitude of the vaporization transient timescale ?

A1.4: The order of magnitude of the vaporization transient timescale is characterized by the time scale in Eq. (15). The ‘vaporization number’ that matches our simulations best was $Va = 0.31$, i.e. the dimensional timescales are of the order as shown in Fig. 3i. For a droplet diameter of 0.001 m, we found the droplet vaporizes in a span of seconds.

R1.5: In figure 3g and h, can you precise the time for each image ?

A1.5: Thank you for the suggestion. The time of each snapshot has been included in the caption.

R1.6: I think the figure 2f has to be more described for an ease of understanding:

A1.6: Thank you for the suggestion. The caption for Fig. 2f has been extended to make the caption and the plot clearer: “The background contour plot shows the maximum spatial density gradient, the solid dots show the formation of stable TGIIF, and the non-filled points represent diffusive interfaces. The colors of the points correspond to the farfield reduced temperatures, consistent with (d) and (e).”

R1.7: The relation 12 and figure 3i should be more detailed, why $Va = 0.31$?

A1.7: Relations (11) and (12) are the closed form temperature distribution equations obtained from a piecewise linear thermal conductivity model. A detailed derivation is found in the methodology, we now refer to it in the text.

Fig. 3i compares the computed 2D droplet lifetimes to timescales obtained from Eq. (15), over a wide range of parameters, i.e. $Tr = [1, 1.5, 2, 2.5]$, $pr = [1.5, 2, 4, 6]$, $D = [0.001, 0.01, 0.1]$ m. Dimensional analysis can only reveal relations up to a proportionality constant, without predicting its value. Here, Va is that proportionality constant, and $Va=0.31$ leads to the best agreement between the computational and the theoretical result when using Eq. (15) to determine the droplet lifetime. In a way, $Va=0.31$ can thus be interpreted as the characteristic Va for the problem at hand, comparable in meaning maybe to a Strouhal number $St = 0.21$ that characterizes the famous von Karman vortex street in cylinder flow. We extended the discussion to make this clearer.

Reviewer 2:

R2.1: The work is devoted to the analysis of supercritical fluids in the presence of temperature gradients inside the system. Both analytical results and results on computer simulation of real fluids were obtained. It is concluded that in the presence of temperature gradients, an analog of surface tension will arise in the system. The equilibrium distribution of the order parameter (eg density deviation from the mean) is given by the temperature distribution, hence its gradient.

A2.1: We would like to thank reviewer R2 for the time and the valuable comments.

R2.2:Hence, surface tension automatically arises. The results themselves are trivial and do not correspond to the level of the journal.

A2.2: The fundamentally surprising result of our work is that surface tension does, in fact, *not* arise, and, moreover, that surface tension is not even necessary for the formation of stable interfaces. This result contradicts more than 120 years of chemical thermodynamics, since van der Waals developed gradient theory in 1894 and predicted that no pure fluid interfaces can be formed at supercritical pressures. These base findings are the foundation of countless theories developed since then, e.g. nucleation theory. We extended the introductory text to emphasize how our results contradict established knowledge. As an additional example, numerous groups are trying to understand and explain droplet formation at fuel supercritical pressures as relevant for Diesel engines, jet engines, or rocket engines – all of which *start* from the premise that this is exclusively possible in a mixture context. Our results show that this is not true and that there may exist a completely different class of explanations based on pure fluid thermophysics instead.

R2.3: Moreover, the situation with temperature gradients is, by definition, nonequilibrium.

A2.3: This is certainly true but does not limit any of our results. The whole fields of heat transfer and fluid mechanics have been developed to deal with nonequilibrium processes.

R2.4: No estimates of the characteristic resorption times for temperature and density gradients are given. If we perform a detailed analysis of the kinetics of processes and characteristic times for real fluids, then the article could be published in a technical journal.

A2.4: The key physical phenomenon acting here is heat transfer. We provide a detailed analysis of associated timescales, introduce a new characteristic timescale, derive a new nondimensional parameter, and demonstrate that it accurately describes relevant time scales over a wide range of parameters ($Tr = [1, 1.5, 2, 2.5]$, $pr = [1.5, 2, 4, 6]$, $D = [0.001, 0.01, 0.1]m$). Our relation for the associated characteristic timescales predicts droplet vaporization times to within 50% over 5 orders of magnitude of timescales (see Fig. 3i).

Reviewer 3:

This paper highlights a very interesting research with is intriguing with unusual conclusions. The conclusion of the authors is explicit and illustrate my purpose. They say: “Ultimately, this work questions and extends the definition of what constitutes a droplet or a bubble”. Yes, this is a good question. The main idea is to consider some cases with a temperature gradient across the Widom line (between supercritical liquid and gaseous states). They call it “thermal gradient induced interfaces (TGIIIF)”. The interface is defined when the density has an inflection point. The mathematical and thermodynamical concepts are classical (Fourier law, simple algebra) even if some parts are derived without some clear explanation. The paper is of good quality even if one the authors is a specialist of this topic as he has been publishing on Widom line for quite a few years. The idea is very interesting even if it is unexpected. I think the paper can be published in “Nature&Communications” after having answered to my following questions/remarks:

A: We would like to thank the reviewer for the assessment that this topic is very interesting and unexpected and that it should be published after addressing the remarks below. We are positive we have addressed each question and remark for the revised manuscript.

R3.1: Is it appropriate to call it “droplet”? We are in fact in liquid and gas like regions but without any surface tension. I am not sure if it is appropriate to call it “droplet”.

A3.1: We admit that calling these structures ‘droplets’ was deliberate, maybe a bit provocative. However, this is an important part of our results: as sketched in Fig. 1a, a ‘true’ droplet exhibits a finite thickness interface and by no means an instantaneous switch between liquid and gas. In that sense, the stabilized interfaces have that same characteristic - it’s a quantitative rather than a qualitative difference. And the big common feature is the stabilized interface that causes regression towards a circular cross section - like in droplets with surface tension. We thus think that emphasizing the similarities is justified. Further, but avoiding delusions of grandeur, one could argue that this work questions and extends the definition of what a droplet is, so we call it a droplet as we extend the definition of a droplet to include TGIIIF formed droplets.

R3.2: The authors state the supercritical vaporization is analogous to subcritical droplet evaporation and in the conclusion they are talking about multiphase interfaces which troubles me a little bit. A question related to these facts: what would be the result of a 2D or 3D modelling? I guess the curvature might play a non-negligible role.

A3.2: Thank you for the comments. Again, we agree that we deliberately emphasize the similarities between 'true' droplets and stabilized interfaces. However, as you state yourself, in either case the (molecular) situation is that of a (closely packed) liquid in the center, surrounded by a (looser) gaseous fluid. In either case, the interface exhibits a gradual transition between liquid and gas. Is this here then not a multiphase situation? One could argue that our thermally stabilized interfaces are not in equilibrium, which is certainly true. However, is a vapor layer in subcritical film boiling of water over a hot surface then not a multiphase interface either? We readily admit that there are not obvious answers to these questions but decided that a 'generalized' view of droplets allows the reader to use existing intuition.

While the first part of the paper deals with 1D steady interface formation, the second part focuses on transient 2D 'droplet' vaporization. Lacking surface tension, the curvature here does not introduce changes in the local thermodynamic pressure. However, the curvature does introduce a quantitative difference in the vaporization time scales (more surface area for heat transfer per enclosed volume leads to faster vaporization). 2D also offers for an expansion and variable geometry, c.f. Fig. 2b, in which an ellipsoidal cross section regresses towards a circular shape through enhanced vaporization at the (higher curvature) 'corners', and simultaneous expansion of the whole droplet structure. Interestingly, we find the same D-square scaling that is classically established for subcritical droplet vaporization by Spalding - however, we arrive from a completely different derivation based on heat transfer rather than an evaporation process. We have a few 3D results which qualitatively show the same behavior as 2D droplets, so we decided they do not add to the discussion. E.g., integration into Fig. 3i would require for more adjustments due to a different surface/volume scaling. This could be worthwhile for a follow-up publication in a more technical journal but we find that this does not provide new insight for this paper.

R3.3: I guess the time effect is important in figure 2 as clearly shown by the authors. The sharp interface scenario does not stay forever. Would one considers that this resembles to a real interface with surface tension? I do not think so.

A3.3: You are of course right that, depending on the set-up, supercritical droplets indeed vaporize and vanish when all of the liquid has been heated to temperatures above the pseudo boiling temperature T_{pb} . However, the interfaces are not intrinsically transient: the first part of the paper shows how the interfaces stabilize in a (eternal) steady state for a given sustained temperature gradient. This is not a hypothetical fringe case: in our recent paper "Onset of heat transfer deterioration caused by pseudo-boiling in CO₂ laminar boundary layers" (International

Journal of Heat and Mass Transfer, Volume 193, 1 September 2022, 122957) we describe how a steady layer of a supercritical gas film forms over a heated wall, just like in many power plants - and incidentally, just like in subcritical heat exchangers with surface tension. Both subcritical and supercritical cases even give rise to the same phenomenon of heat transfer deterioration! Concerning droplets, we have to note that subcritical droplets with surface tension will eventually evaporate likewise, such that the interface vanishes when all of the liquid is transformed, just like in our transient droplets in Fig. 2.

R3.4: Eqs. (1-7) are clear but the explanation which follows on page 5 is not easy to follow as some parameters are not clearly defined for the transcritical case. What does mean “pb”? How to define k_{min} ? What is the value of T^* ? This part needs to be more developed.

A3.4: Thank you for the comments. We have updated the manuscript to include an explanation of the variables used: “ T_{pb} is the pseudo boiling temperature which characterizes the supercritical liquid--gas transition and is defined as the locus of the isobaric heat capacity peaks. k_{min} is the minimum value of the thermal conductivity evaluated at constant pressure and T^* is the temperature at which k_{min} is reached.”

R3.5: There is a mistake on page 4 and 5 (Eq. (14) should be Eq. (4)).

A3.5: Thank you for catching that mistake, we updated the manuscript accordingly.

REVIEWER COMMENTS

Reviewer #1 (Remarks to the Author):

I am satisfied with the answers provided by the authors. Accept as it is.

Reviewer #2 (Remarks to the Author):

I was able to understand what was done in this work.

1) If there is a strong temperature gradient in some area of the fluid, it will cause density gradient is trivial.

2) The temperature gradient in the supercritical region after injection is rapidly will resolve. The authors do not want to give times, although in response to the reviewer 1 talking about some seconds! Where do these seconds come from!?

3) And here the line of pseudo-boiling (Widom line) I did not understand at all. Such lines near the crit point - hundreds - maxima of heat capacity, compressibility, thermal expansion, density fluctuations, etc. And they are all different! And the lines in general of all the highs

will be different depending on the path on the phase diagram - along isotherms, isochore, isobar, etc. The authors took some one line of Cp maxima on the isotherms. So what?

Why do they say that there will be some big gradients on this line?

Moreover, all Widom lines disappear very quickly when moving away from the crit point

(the maxima decrease and blur). And the authors themselves say that their

the mechanism works even very far from the crit point

I don't understand anything

Reviewer #3 (Remarks to the Author):

Even if the definition of an interface is not classical, the authors have tried to justify in their answers to my questions. Yes the purpose of the paper is intriguing but highly interesting as I said in my first report.

I accept the publication of the paper in the present form.

Response to Reviewers

We would like to thank the reviewers for the time and the valuable comments. We are excited to read that R1 “really appreciated this work” and finds “that this contribution is quite original”, and that R2 finds “the idea is very interesting” and “unexpected”. Without doubt, addressing the comments has helped make the paper much clearer. Thank you!

Reviewer 1:

I am satisfied with the answers provided by the authors. Accept as it is.

A: We would like to thank Reviewer 1 for the time, comments, and the support to publish this manuscript!

Reviewer 2:

R2.1: I was able to understand what was done in this work. If there is a strong temperature gradient in some area of the fluid, it will cause density gradient is trivial.

A2.1: We couldn't agree more, the formation of a diffusing density gradient is indeed what we expected, e.g. the result for an ideal gas in Fig. 1g.

What we did not expect, however, and what defies any intuition based on the known laws of diffusion, is that we found conditions at which density gradients *increase over time*, with no force acting, see Fig. 2.

We added supplementary movies of a droplet with an initially diffuse interface *sharpening* to illustrate this further - it looks like the movie is running backwards. We added three movies in total,

1. `diffuse_drop` shows the expected behavior of a cold blob just diffusing outwards and disappearing;
2. `sharp_drop` shows the unexpected behavior of interface sharpening and then contracting during vaporization;
3. `interface_sharpening` shows the process of interface sharpening in more detail.

Imagine a droplet of ink suspended in water, and rather than diffusing outwards (reducing the concentration gradients), the ink drop contracts and sharpens its interface ink/water instead! This is what we found for the density field under certain conditions. We rewrote the opening paragraph to make this clearer.

In a steady analysis, an interface with a density inflection point forms and persists, in absence of any acting forces or a phase equilibrium. This inflection point is very much an unexpected phenomenon, previously only observed in liquid-vapor equilibrium: van der Waals used his equation of state and introduced his gradient theory more than a century ago in which he showed that under liquid-vapor equilibrium, a density inflection point establishes at the liquid-vapor interface.

Now, what we show here is that we can identify thermodynamic conditions in which a density inflection point, i.e. an interface, exists even under conditions without liquid-vapor equilibrium - indeed, at conditions at supercritical pressures where a liquid-vapor equilibrium cannot even exist, c.f. Fig. 1h.

The key insight is not the density gradient, it is *sharpening* of the density gradient and the formation of a gradient *inflection point*, the persistence of a localized large value of the density gradient embedded in areas of low density gradient, something our intuition and diffusion laws tell us should be smoothed out by diffusion - but surprisingly, it's not!

R2.2: The temperature gradient in the supercritical region after injection is rapidly will resolve. The authors do not want to give times, although in response to the reviewer 1 talking about some seconds! Where do these seconds come from!?.

A2.2: We're happy to give times! We would like to bring Fig. 3i to your attention in which we compile dimensional vaporization times for a wide range of evaluated conditions (different diameters, pressures, ambient temperatures).

The computed droplet lifetime on the y-axis of Fig. 3i was determined by running CFD simulations of a cold 'droplet' suspended in a warm environment at various conditions (droplet diameter, ambient temperature, pressure). The interface was determined as the density inflection point which allows to define a droplet radius. The droplet lifetime is then the time in the simulation at which that radius vanished.

The characteristic vaporization time t on the x-axes of Fig. 3i was determined using Eq. (14) which we obtained from dimensional analysis, see Sec. 4.5. The data needed for Eq. (14) are initial conditions (diameter and temperature difference) and tabulated fluid properties (minimum thermal diffusivity and the temperature at which it occurs, both functions of the pressure). V_a is then a nondimensional proportionality constant that cannot be obtained from dimensional analysis. For $V_a=0.31$ we find best agreement between droplet lifetimes determined from CFD and Eq. (14) over a wide range of conditions and several decades of predicted droplet lifetimes (from ~5s to ~10000s).

We retitled Section 4.5 in the methods section, “Time scale analysis”, to make clearer where our timescale is developed in the paper.

R2.3: And here the line of pseudo-boiling (Widom line) I did not understand at all. Such lines near the crit point - hundreds - maxima of heat capacity, compressibility, thermal expansion, density fluctuations, etc. And they are all different! And the lines in general of all the highs will be different depending on the path on the phase diagram - along isotherms, isochore, isobar, etc. The authors took some one line of Cp maxima on the isotherms. So what? Why do they say that there will be some big gradients on this line? Moreover, all Widom lines disappear very quickly when moving away from the crit point (the maxima decrease and blur). And the authors themselves say that their the mechanism works even very far from the crit point.

A2.3: We agree that using “the” Widom line introduces a lot of ambiguity, because its definition as maximum of thermodynamic response functions - as you say! - allows for *hundreds* of different lines, instead of a single one. The pseudo boiling line resolves this ambiguity through a precise definition with thermodynamic meaning: the maximum isobaric curvature of the free Gibbs enthalpy is a supercritical generalization of the subcritical discontinuous change in slope observed at a phase transition. It can be approximated by the maxima in cp/T or, close to the critical point, by maxima in cp . In that sense it is related to Widom lines, but without the ambiguity.

The pseudo boiling line furthermore approximately marks the steepest isobaric thermal density gradient, i.e. an inflection point in the density upon isobaric heating. In presence of a spatial temperature gradient - as is often the case for injection with cool droplets in hot environments (e.g. in rocket engines, jet engines, Diesel engines) - the pseudo boiling line thus seems like a likely candidate to cause a *spatial* density inflection point when the maximum thermal density gradient $(d\rho/dT)_{\max}$ [a fluid property] coincides with a spatial temperature gradient (dT/dx) [a process property], such that $(d\rho/dx)_{\max} \sim \{(d\rho/dT)(dT/dx)\}_{\max}$.

We added a paragraph to the introduction to make this clearer:

“However, the question arises as to whether the aforementioned pseudo boiling could provide a mechanism for droplets to form under true supercritical conditions, i.e. in absence of a

phase equilibrium. In pseudo boiling, the liquid-gas transition occurs over a finite temperature interval across the pseudo-boiling-line \cite{BanutiJSF2020}, an extension of the coexistence line. It is related to the Widom line \cite{Simeoni2010,Xu2005,Sciortino1997,BU2003} which is defined as locus of extrema in the thermodynamic response functions and thus can have many different characteristics depending on which response function is chosen \cite{BanutiJSF2020}. The pseudo boiling line resolves this ambiguity with a precise definition based on the curvature of the free Gibbs enthalpy \cite{BanutiJSF2020}. It is a fluid property and approximately marks the steepest isobaric thermal density gradient $(\partial \rho / \partial T)_p$. In many injection problems, a spatial temperature gradient $(\partial T / \partial x)$ exists between the droplet and its surroundings \cite{Mayer1998, Manin2014, Crua2017, Delplanque1993, Yang1994, Sirignano1999, Qiu2015, Ma2019}. Thus, the pseudo boiling line seems a likely candidate to induce a maximum in the spatial density gradient $(\partial \rho / \partial x)$, i.e. a density inflection point, in the presence of a spatial temperature gradient -- much like what we see in subcritical droplets \cite{DahmsPCI2015}."

R2.3: I don't understand anything.

A2.4: Thank you for your questions, I hope we could clear things up better!

Reviewer 3:

Even if the definition of an interface is not classical, the authors have tried to justify in their answers to my questions. Yes the purpose of the paper is intriguing but highly interesting as I said in my first report. I accept the publication of the paper in the present form.

A: We would like to thank Reviewer 3 for the time, questions, comments, and the support to publish this manuscript!

REVIEWERS' COMMENTS

Reviewer #2 (Remarks to the Author):

Unfortunately, I did not fully understand the answers and arguments of the authors. I readily admit that this is my problem. Since 2 other reviewers are positive, I don't feel confident to reject the article. Let it be published. Readers will judge.

Response to Reviewers

Reviewer 2:

R2.1: Unfortunately, I did not fully understand the answers and arguments of the authors. I readily admit that this is my problem. Since 2 other reviewers are positive, I don't feel confident to reject the article. Let it be published. Readers will judge

A2.1: We thank you for the time and valuable comments as your comments have helped make the paper much clearer. Thank you!